# Rescuing Infected Deep Brain Stimulation Therapies in Severely Affected Patients

**DOI:** 10.3390/brainsci13121650

**Published:** 2023-11-28

**Authors:** Thomas Fortmann, Samer Zawy Alsofy, Marc Lewitz, Antonio Santacroce, Heinz Welzel Saravia, Ioanna Sakellaropoulou, Eike Wilbers, Steffen Grabowski, Ralf Stroop, Zafer Cinibulak, Makoto Nakamura, Ralph Lehrke

**Affiliations:** 1Department of Medicine, Faculty of Health, Witten/Herdecke University, 58448 Witten, Germany; szawyalsofy@barbaraklinik.de (S.Z.A.); mlewitz@barbaraklinik.de (M.L.); antoniosantacroce@msn.com (A.S.); ewilbers@barbaraklinik.de (E.W.); ralf@stroop.de (R.S.); cinibulakz@kliniken-koeln.de (Z.C.); nakamuram@kliniken-koeln.de (M.N.); 2Department of Neurosurgery, St. Barbara-Hospital, Academic Hospital of Westfaelische Wilhelms-University Muenster, 59073 Hamm, Germany; hwelzel@barbaraklinik.de (H.W.S.); isakellaropoulou@barbaraklinik.de (I.S.); stgrabowski@barbaraklinik.de (S.G.); 3Department of Stereotactic Neurosurgery, St. Barbara-Hospital, Academic Hospital of Westfaelische Wilhelms-University Muenster, 59073 Hamm, Germany; rlehrke@barbaraklinik.de; 4European Radiosurgery Center Munich, 81377 Munich, Germany; 5Department of Neurosurgery, Academic Hospital Koeln-Merheim, Witten/Herdecke University, 51109 Koeln, Germany

**Keywords:** deep brain stimulation, infection, salvage surgery

## Abstract

(1) Background: Infections in deep brain stimulation (DBS) hardware, while an undesired complication of DBS surgeries, can be effectively addressed. Minor infections are typically treated with wound revision and IV antibiotics. However, when visible hardware infection occurs, most centers opt for complete removal, leaving the patient in a preoperative state and necessitating post-removal care. To avoid the need for such care, a novel technique was developed. (2) Methods: The electrodes are placed at the exact same spot and then led to the contralateral side. new extensions and a new generator contralateral to the infection as well. Subsequently, the infected system is removed. This case series includes six patients. (3) Results: The average duration of DBS system implantation before the second surgery was 272 days. Only one system had to be removed after 18 months due to reoccurring infection; the others remained unaffected. Laboratory alterations and pathogens were identified in only half of the patients. (4) Conclusions: The described surgical technique proves to be safe, well tolerated, and serves as a viable alternative to complete system removal. Importantly, it effectively prevents the need of post-removal care for patients.

## 1. Introduction

Infections of deep brain stimulation (DBS) hardware are an unwanted but possible complication of DBS surgeries. The literature reports infection rates from between two percent up to fifteen or even twenty percent in older case series [1,2,3,4,5,6,7,8,9,10,11,12,13]. Infections of implants are always problematic for patients as well as for their treating physicians. Several risk factors for infection after deep brain stimulation are mentioned in the literature: age, body mass index, procedure side, gender, smoking, diabetes mellitus, operating surgeon, scalp erosion, duration of surgery, number of individuals in the operating room, season, a diagnosis of Parkinson’s disease, dystonia, essential tremor, and newer indications for DBS [1,14,15,16,17,18,19,20,21]. Interestingly, some studies show higher risks for patients with Parkinson’s disease, some for patients with essential tremor, but most for patients with dystonia. For minor infections, it might be sufficient to treat a patient with intravenous (IV) antibiotics or to perform wound revision in combination with IV antibiotic treatment [22,23,24]. If wounds are purulent or if hardware is perforating the skin, it is likely that a biofilm is present upon the hardware. Biofilms of bacteria on the implants are the reason for reoccurring wound healing difficulties [25]. Removing necessary implants—even partially—results in the reoccurrence of the treated disease symptoms (akinesis, tremor, or dystonia). Patients that are benefiting from their DBS therapies are keen to retain their therapy. Almost all centers remove the infected hardware and treat the patient with targeted IV antibiotics, thereby often leaving the patient in a condition worse than the preoperative state due to the natural progression of their diseases [10,24,26,27,28,29,30]. This is especially true for patients with severe cases of dystonia, Parkinson’s disease, and tremor, who are profiting immensely from their DBS therapies. For patients with Parkinson’s disease, it is possible to partially compensate for the loss of stimulation with L-dopa medication, but patients with tremor are often left with such severe shaking that they require nursing care [31,32]. Since medication does not suppress their tremors efficiently, they lose their autonomy. They may be unable to feed or wash themselves, therefore requiring full-time nursing care within a nursing home. Patients with dystonia might even suffer from a dystonic crisis that requires intensive care treatment.

Since most infections are located either at the generator or at the connector, the scars above the electrodes are (generally) unaffected [33,34,35]. Implant infections are often low-grade infections of low-virulence bacteria that can be treated by removing the infected implants [36]. Taking this into consideration, we hypothesized that it is possible to maintain the DBS therapy and the patient’s autonomy by placing new electrodes leading them to the contralateral side. New extensions and a new generator are also placed contralateral to the infected side. Subsequently, the infected system is removed, and the wounds are cleaned and taken care of. Patients are then treated with IV antibiotics. Due to our lengthy experience with deep brain stimulation (since 1996) and about 40 new DBS successful implantations every year, we were confident in offering patients an alternative to removing the DBS therapy. Searches of PubMed and MEDLINE conducted on 2 October 2023 (“deep brain stimulation AND infection”) found only one publication with a similar approach: the implantation of a new system while the infected system is removed in the same session [37].

Here, we describe a case series of six patients that were treated using this technique. We aimed to determine whether this procedure is a safe alternative to removing the system and waiting three months before reimplantation.

## 2. Materials and Methods

All six patients suffered from severe essential tremor. After treatment of their tremor with oral medication failed, they underwent their initial DBS surgeries targeting the ventral intermediate nucleus of the thalamus (VIM). They all received single-shot antibiotic treatment before their surgeries, with the IV treatment repeated every three hours throughout the surgery. The scalp was washed with an octenisept solution and hair was shaved only where the planned skin cuts were. The outcome was beneficial for all six patients; all regained their autonomy. All presented with an infection of their implants retroauricular or at the side of their generator. When their stimulation was switched off to see the severity of their tremor in case of hardware removal, all patients were severely affected. The patients were offered the alternative treatment but were informed that there was little to no experience regarding the outcome of this new procedure. Informed consent was obtained for all six patients. On the day of the procedure, the patients received total intravenous anesthesia and a single shot of 1.5 g cefuroxime. The patient’s head was fixed in a ceramic ring (Precisis AG, Heidelberg, Germany) within a stereotactic X-ray setup (MTS, Marl, Germany). Two planar X-ray pictures were taken and uploaded into the inomed iPS 6.0 software (inomed, Emmendingen, Germany, Figure 1A). The tips and entry points of the electrodes were tagged within two planar stereotactic X-ray pictures (Figure 1B) and fused with an empty CT scan dataset of the ceramic ring with a localizer to obtain stereotactic coordinates and trajectories (Figure 1C). The unaffected frontal scars were shaved, washed with the octenisept solution, draped, reopened, and the entry points for the electrodes were exposed. The electrodes were cut as distally as possible, and the Palacos fillings were removed. The security sutures were then opened. Meanwhile, the stereotactic coordinates were set on a Riechert–Mundinger frame (Precisis AG, Heidelberg, Germany) and verified using the stereotactic phantom (Precisis AG, Heidelberg, Germany). The frame was set up at the ring and the implanted electrode was removed. Before removing the electrode, the skull level was marked on it. The distance to the tip was measured and transferred to the new electrode. The new electrode was placed via the existing pathway while checking the correct path of the electrode with intermittent X-rays (Figure 1D). The correct position was confirmed by stereotactic X-ray, and the electrode was secured with a suture and a microplate. The final position was again confirmed by stereotactic X-ray (Figure 1E). All electrodes followed their previous trajectories. In the case this did not occur, new trajectories through new burr holes were already planned. After fixing the electrodes, they were guided toward the contralateral side. The frontal wounds were then closed, followed by the temporary closure of the retroauricular wound. The ring fixation was removed, and the head rotated towards the infected side. New extensions and a new pacemaker were placed contralateral to the infected side, and the wounds finally closed. Now, the head was rotated away from the infection. After new sterile washing and covering, the thoracic and retroauricular wounds were opened. The pacemaker was removed, and the extensions, along with the remaining electrode connected to the extensions, were removed through the retroauricular wound. After cleansing the wounds and excising avital wound margins, the wounds were finally closed.

## 3. Results

### 3.1. Patients’ Charateristics

Six patients, all diagnosed with severe essential tremor on their dominant body side, had implants exposed retroauricularly or thoracically, and unaffected frontal scars. They were offered this procedure because they would require nursing care in case of complete implant removal.

Patient 1 presented at age 81 with an 8-year history of a progressing essential tremor, particularly affecting her dominant left side. Various medications were attempted, but none effectively alleviated the tremor. Due to her inability to feed herself, she underwent VIM stimulation, with a generator implanted via a submammarian approach in 2012 [38], regaining autonomy. The generator was replaced in 2015 and 2018 using the same approach. In 2020, at the age of 89, she complained of pain in the right breast. A dislocation of the generator was diagnosed, threatening to perforate the skin medial to the nipple. In March 2020, the generator was mobilized and placed beneath the clavicle through a para-axillary approach. The thinned-out skin above the generator was excised, and wounds were properly closed. In May 2020, the patient presented with retroauricular wound dehiscence. We admitted the patient on that Friday, planning the salvage surgery for Monday. On Sunday, the wound at the breast opened, emptying pus (Figure 2). The salvage surgery was performed on Monday as planned.

Patient 2 began experiencing essential tremors during adolescence. Various medications effectively controlled the tremor for an extended period. However, when the patient was 76 years old, the medication failed to control the tremor, especially in his dominant right arm. Writing and eating became extremely strenuous, leading to the patient’s social isolation due to shame. In January 2018 at the age of 78, he received deep brain stimulation, regaining autonomy. However, in February 2019, while combing his hair, the patient suddenly experienced shaking again. Upon inspection by his wife, a torn cable sticking out of the skin was found. The patient was admitted; due to one torn extension and the visibility of the second extension, salvage surgery was performed the following day.

Patient 3 first noticed a tremor in his dominant right hand at the age of 43 when his wife was diagnosed with breast cancer. Seven years later, after his wife’s death, the tremor extended to the left side and his head. Several medications initially alleviated the tremor, but the patient withdrew socially, and the tremor progressed beyond the control of medication. Before losing autonomy, he underwent VIM implantation in September 2010. In 2011, the left-hand tremor reoccurred, and high impedance on the right side was detected. X-rays revealed Twiddler’s Syndrome with a broken right electrode. A new DBS system was implanted in February 2011. In June 2013, a generator replacement took place, and revisions in July and August addressed minor infections. In May 2014, the patient was readmitted with a purulent infection above the generator, leading to salvage surgery the following day. Meanwhile, the patient developed severe chronic heart failure due to aortic valve stenosis. Because the patient was unable to undergo a cardiothoracic procedure, no further generator replacements took place. Unfortunately, the patient also experienced skin perforation of the right electrode due to wearing overly tight-fitting caps (Figure 3). The electrode was removed under local anesthesia. During the coronavirus crisis, the patient developed oxygen dependence with obstructive sleep apnea syndrome (OSAS). Due to wearing a mask for OSAS, he developed skin perforation of the connectors behind his left ear, which lead to the removal of the extensions under local anesthesia.

Patient 4 first noticed a tremor at the age of 25. Propranolol effectively treated the tremor in his dominant right arm for many years. However, around the year 2000, the tremor worsened, affecting both upper extremities. Various medications were tried without success, and daily activities such as dressing and eating became troublesome. In 2005, he underwent VIM-DBS therapy, regaining autonomy. In 2013, the generator was replaced. In April 2015, a bee sting above the generator became infected, secreting pus. Shortly afterward, the generator pocket became swollen, red, and painful. The patient opted against surgical intervention, choosing a four-week course of IV antibiotic treatment. The symptoms resolved, leading to cessation of treatment. In May 2015, signs of infection reappeared above the generator (Figure 4) and the retroauricular wound. Antibiotic treatment was resumed, managing the infection until September 2015, when a wound dehiscence was diagnosed above the generator. The patient was admitted, and upon switching off the stimulation, severe tremors of both upper and lower extremities and the head were observed. The salvage procedure was performed the following day.

Patient 5 experienced a slight tremor in her left, dominant hand at the age of 40. Initially, she received different medications to alleviate the tremor. However, as the tremor worsened, affecting her right hand, both legs, and the head, medication became ineffective, and she required assistance from her children. At the age of 70, in 2009, she underwent VIM stimulation, and the generator was implanted via a submammarian approach, leading to the regaining of full autonomy. In December 2012, the generator was replaced. However, in January 2013, the patient developed intermittent galactorrhea due to a fixed 90-degree rotation of the generator along its longitudinal axis. In July 2013, a wound revision with generator repositioning was performed. In December 2015, another generator replacement was carried out. Following this surgery, the submammarian wound became infected, leading to salvage surgery performed thirteen days later.

Patient 6 was diagnosed with essential tremor in childhood, which affected both upper extremities and progressed slowly. Medication successfully managed the symptoms for a long time. However, as the patient became socially isolated and was no longer able to care of her household, she underwent VIM stimulation in June 2011 at the age of 68. In January 2016, the generator was replaced. Due to caudal dislocation and subsequent painful tension from the extensions, the generator was reattached, and the extensions freed from scar tissue. Despite these efforts, the extensions continued to cause pain, leading to their replacement. Following this surgery, the wound above the generator exhibited early signs of healing problems, and IV antibiotics were administered. Then, 32 days later, the patient was readmitted with pus draining from the wound. Salvage surgery was performed the following day.

Six patients, comprising three women and three men, with an average age of 81.6 years (ranging from 73 to 89 years) and severe essential tremor, were offered this procedure (Table 1). Five patients developed thoracic wound infections, while one experienced a retroauricular infection (Patient 2). On average, the deep brain stimulation system was implanted 272 days before the salvage surgery (ranging from 13 to 805 days). As of today (starting from 2014), the salvage surgeries were successful for four patients, with an average follow-up of 58.0 months (ranging from 18 to 89 months). Patient 2 had a purulent infection 18 months after the salvage surgery, leading to the explantation of the entire hardware. Three months later, a new VIM stimulation was performed. Patient 3 presented 89 months after the salvage surgery with an aseptic skin perforation on the right electrode. Due to cardiac comorbidities, general anesthesia and supine position were not possible. Therefore, the electrode was removed under local anesthesia in a sitting position.

### 3.2. Patients’ Comorbities

All patients had comorbidities, ranging from two to six (Table 2). On average, the patients had a body mass index of 30 kg/m^2^ (ranging from 20.9 to 36.7 kg/m^2^). Regarding the comorbidities, five patients had a history of arterial hypertension, three had a history of cardiac disease (coronary artery disease, atrial fibrillation, dilatative cardiomyopathy, aortic valve stenosis, bicuspid valve stenosis), two had hypothyroidism, one had dementia, one had non-insulin-dependent diabetes mellitus type 2, one had chronic obstructive pulmonary disease, one had a history of breast cancer, one underwent cholecystectomy, one had obstructive sleep apnea syndrome, and one experienced nicotine abuse.

### 3.3. Infection Charateristics

Three patients exhibited signs of infection in their blood work (Table 2), manifesting elevated C-reactive protein values (4.6–9.3 mg/dL, normal <0.5 mg/dL). One patient had an increased leukocyte count (14,000/µL). Given that this patient is a smoker, it remains unclear whether the elevation is attributed to smoking or the infection. Wound smears were taken from all patients, but only in three cases could bacteria be cultivated. These infections were attributed to *Staphylococcus aureus*, *Proteus mirabilis*, and a combined infection with *Staphylococcus aureus* and *Staphylococcus epidermidis*. All patients were treated with IV antibiotics for an average of 8.5 days (ranging from 4 to 14 days, using cefuroxime or flucloxacillin) and, additionally, orally for an average of 6 days (ranging from 3 to 8 days, using cefuroxime, flucloxacillin, cotrimoxazole or rifampicin).

## 4. Discussion

In our case series, all six surgeries succeeded for at least 18 months before an infection occurred in Patient 2. After 89 months of successful treatment for Patient 3, one electrode needed removal due to an avital skin perforation. For four out of six patients, we successfully maintained their therapies until the current day. Is this the right approach to treating a patient in case of an infection? Since there are no clear guidelines indicating when to remove a system or when to treat it with IV antibiotics, we conducted a PubMed and MEDLINE research. When searching for ‘DBS AND infection’, we found 1575 publications. Unfortunately, most publications were unrelated to deep brain stimulation. Therefore, when searching for ‘deep brain stimulation AND infection’, we found 576 publications. Most of these publications focused on indications for deep brain stimulation, with only 42 publications addressing infected hardware (Figure 5). Among these publications, most concentrated on identifying risk factors for infections to prevent them [1,14,15,16,17,18,19,20,21,39,40]. For cardiac pacemakers und spinal cord stimulation devices, the infection rates are similar, ranging between 2 and 3%, as in current DBS studies. The recommended approach for infection is the removal of the infected parts [41,42,43].

With our established treatment protocol involving head washing with an octenisept solution, a single-shot antibiotic treatment of 1.5 g cefuroxime, and only targeted shaving, when necessary, we maintained infection rates between 2 und 3% over the past decade. In instances of minor skin infections, our approach involves wound cleaning and IV antibiotic treatment, aiming to preserve the DBS system. Notably, we have not encountered intracranial infection so far. Given that the majority of infections are traced back to skin flora bacteria, our primary antibiotic of choice is cefuroxime. Vancomycin is reserved as a last-line antibiotic, typically employed in cases of intracerebral infections or those involving Methicillin-resistant *Staphylococcus aureus* (MRSA). An analysis by Tabaja et al. at the Mayo Clinic, examining 73 DBS hardware infections, revealed laboratory results similar to our cohort [44]. They observed CRP elevations in only 59% of patients and abnormal leukocyte counts in 36% of infections. This could be attributed to localized, low-grade infections near the surgical wound, causing early abnormalities in the wound compared to diffuse infections within the skin. Muehlhofer et al. demonstrated that low-grade infections of orthopedic implants often exhibit no significant laboratory alterations due to the presence of low-virulent bacteria [36]. Interestingly, Tabaja et al. reported positive microbiological findings in 97% of cases, potentially influenced by their sampling method. Unlike their approach, we conducted a single-smear test. Enhancing efficiency might involve taking multiple smears and sending samples of infected tissue to the microbiology department.

A study by Chen et al. delved into the costs associated with minor salvage surgeries and the reimplantation of a new system. Their series reported that the patients underwent an average of 2.5 ± 1.4 salvage surgeries. For a patient undergoing the median number of revisions (n = 2), device explantation, and subsequent reimplantation after infection clearance, the mean total cost was USD 75,505. Notably, the recurrence rate of infections after minor salvage surgeries ranged between 27 and 40% [3,22]. In our procedure, the costs of reimplantation are inherently included in the salvage surgery. In cases where patients require nursing care after hardware removal, these costs need to be considered as well. By maintaining patients’ autonomy, the costs of full-time nursing for three months can be avoided. In Germany, the average costs for nursing care range between EUR 63 and EUR 92, depending on the care needed [38]. Thus, requiring care for three months results in costs between EUR 5.292 and EUR 7.728 for nursing, in addition to approximately EUR 35 per day for accommodation and catering (EUR 2940). This represents a distinct advantage of our method compared to the approach of removing an infected system. Most patients vividly remember the severity of their condition before implantation and are motivated to fight for continuation of their therapies. While having a satisfied patient is gratifying for the physician, the decision to treat differently must be based on acceptable risks. Helmers et al. presented a series of six patients treated in a similar manner, providing support for the notion that this treatment approach is a viable alternative [37]. However, what are the potential risks? One concern could be that implanting a new system during an ongoing infection might increase the risk of infecting the new hardware. In our series, only Patient 2 experienced a purulent infection 18 months after the salvage surgery, again with *Proteus mirabilis*. Initially, after the salvage surgery, the wounds were healing properly, and the patient exhibited no signs of a urinary tract infection. The cultivation of *Proteus mirabilis* was then considered a contamination by our microbiologists, leading to the cessation of IV antibiotic treatment after four days. However, with a subsequent purulent *Proteus mirabilis* infection 18 months later, a low-grade infection was considered, necessitating the removal of the system. Even in this patient’s case, who subsequently underwent a new implantation, there was gratitude for having explored this option. We recognize that cefuroxime might not have been the optimal treatment for *Proteus mirabilis*. Moving forward, we plan to critically review whether antibiotic treatment should be stopped on recommendation of the microbiologists or continued for at least seven days.

Given that infections of DBS therapies are primarily focal and spread along the course of the cables, none of our patients developed sepsis. In the event of sepsis, implanting a new hardware system would be avoided due to concerns about potential colonization.

Tanaka et al., in their 2020 paper, demonstrated the possibility of preserving electrodes even after wound dehiscence, suggesting that it may be feasible to salvage DBS therapies even when a biofilm is suspected on the hardware [45]. The success of our procedure has been sustained in four out of six patients until the present day. The first patient underwent this procedure 82 months ago and experienced one uneventful battery change in that period. The duration of antibiotic treatment varied among our patients, ranging from 4 to 21 days, with an average of 8.5 days of intravenous antibiotics. The bacteria causing the infections were mostly, except for Patient 2, bacteria from the normal skin flora, consistent with findings in other publications [2,3,11,12,40,41,46]. Tabaja et al., in their publication, reported treating infections with cefazolin, cefuroxime, or vancomycin, which were used as single-shot antibiotics in the initial surgery as well. However, their study did not demonstrate the same benefit of local treatment of the generator pocket during the initial implantation, as advocated by Abode. Despite this, they recommend a 7–14 day IV antibiotic treatment for minor skin infections and a 14-day IV antibiotic treatment for deep skin infections related to DBS implants [3,45].

Most of our patients were obese, with an average BMI of 30, and had a history of arterial hypertension and other cardiovascular diseases. Obesity has been previously described as a risk factor for infections [45], while arterial hypertension and cardiovascular diseases are not yet extensively discussed in the literature. However, our case series is too small to draw any definitive conclusions, and the observed associations may be influenced by the rising incidence of arterial hypertension and cardiovascular diseases.

With our case series, we present an alternative treatment option to simply removing the implants. We provide a detailed description of the procedure using stereotactic X-ray imaging, making it the first publication of its kind. All patients express gratitude for this option. Apart from preventing the patients from needing nursing care, this procedure also leads to cost savings for both the patients and the social system.

## 5. Conclusions

Implanting a new DBS system on the contralateral side of the infected system, following a strict sequence of surgical steps, proved to be a safe alternative to complete system removal when coupled with adequate postoperative antibiotic treatment. This approach preserves autonomy and, consequently, the quality of life for our patients. Additionally, substantial cost savings were realized, as the patients did not require 24-h full-time nursing care.

## Figures and Tables

**Figure 1 brainsci-13-01650-f001:**
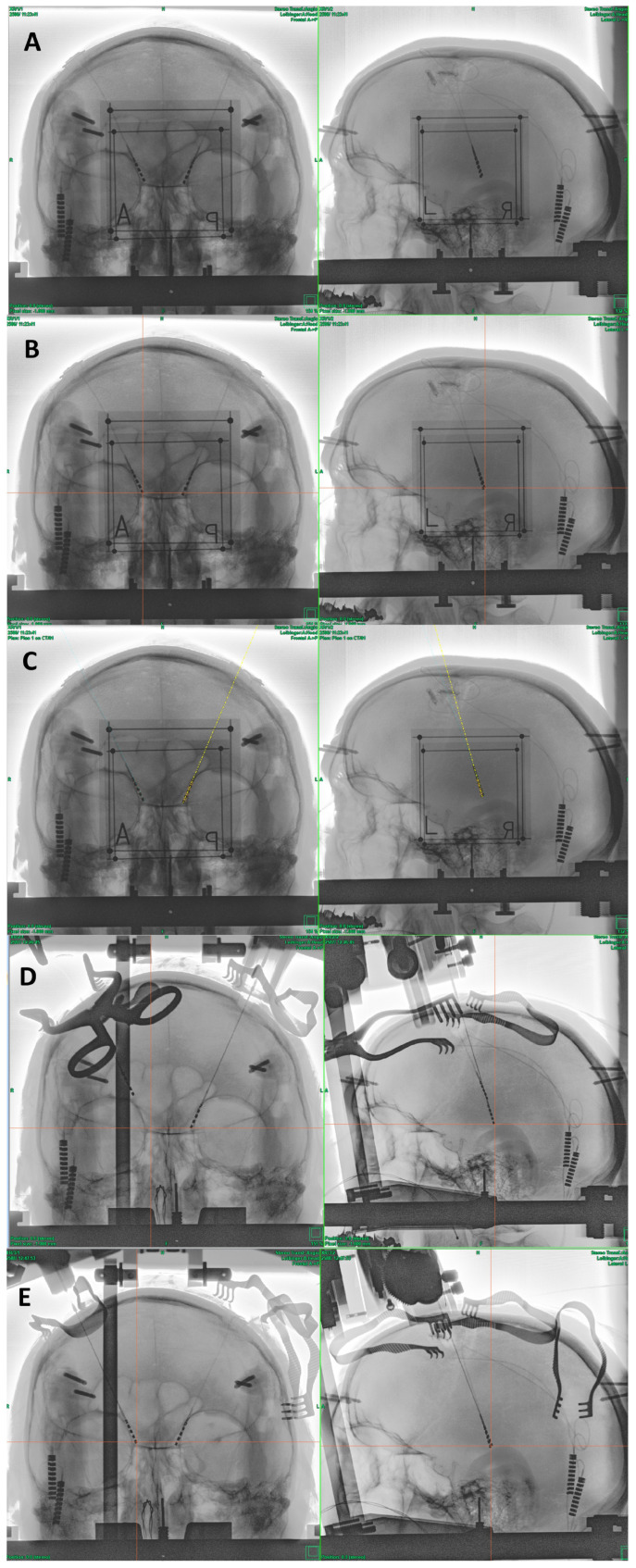
Step-by-step documentation of placing the new electrodes. (**A**) The patient’s head is fixed inside the ring, and two planar X-rays are taken with localizers. (**B**) After fusing an empty CT scan with ring and localizer with the X-rays, the tips and entry points of the electrodes in both X-rays are tagged. (**C**) These coordinates are used to generate the trajectories. (**D**) After removing the old electrode, a new one is inserted into the old pathway. It is documented that the electrode follows the old trajectory. (**E**) After reaching the target point, the electrode is fixed.

**Figure 2 brainsci-13-01650-f002:**
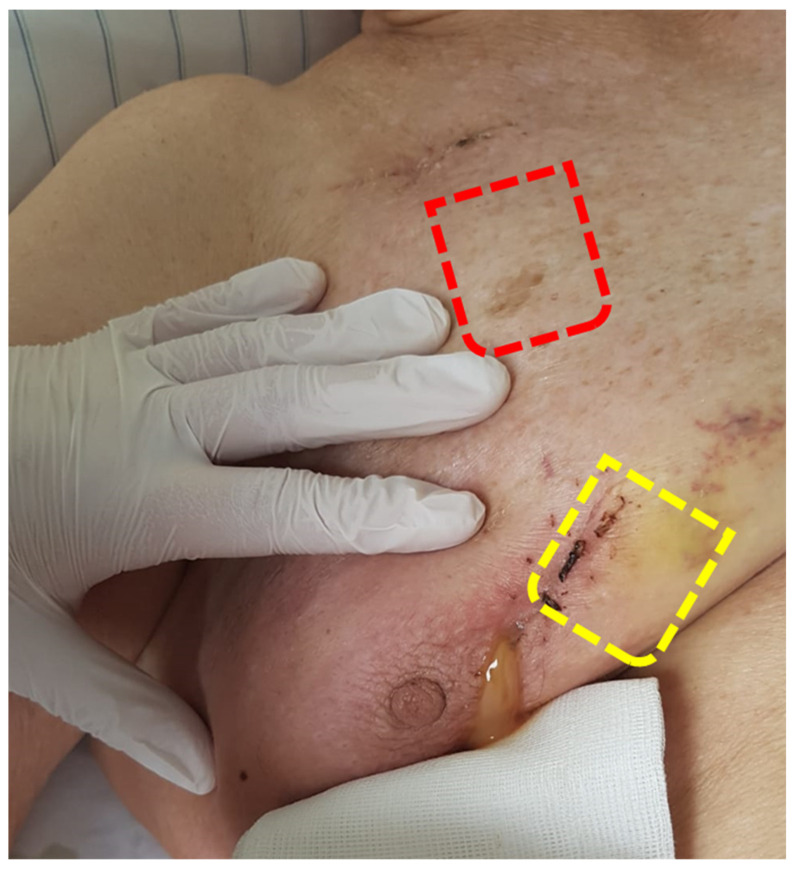
Patient 1, one day before salvage surgery. The red line indicates the current position of the generator, while the yellow line represents its former position, threatening to perforate the skin.

**Figure 3 brainsci-13-01650-f003:**
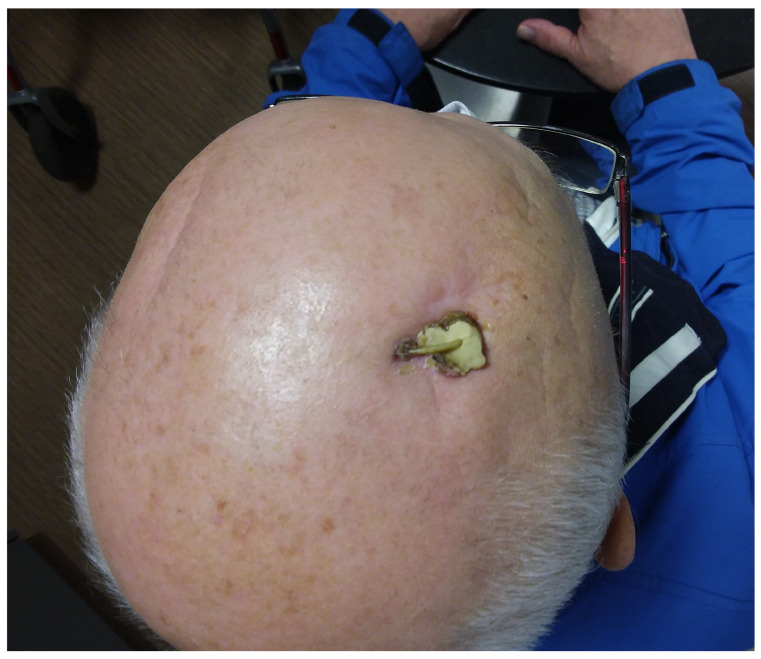
The image shows the condition 89 months after the single-stage replacement surgery for Patient 3. The right electrode was removed, rendering the DBS therapy insufficient. Replacing the electrode was not an option due to the patient’s heart condition.

**Figure 4 brainsci-13-01650-f004:**
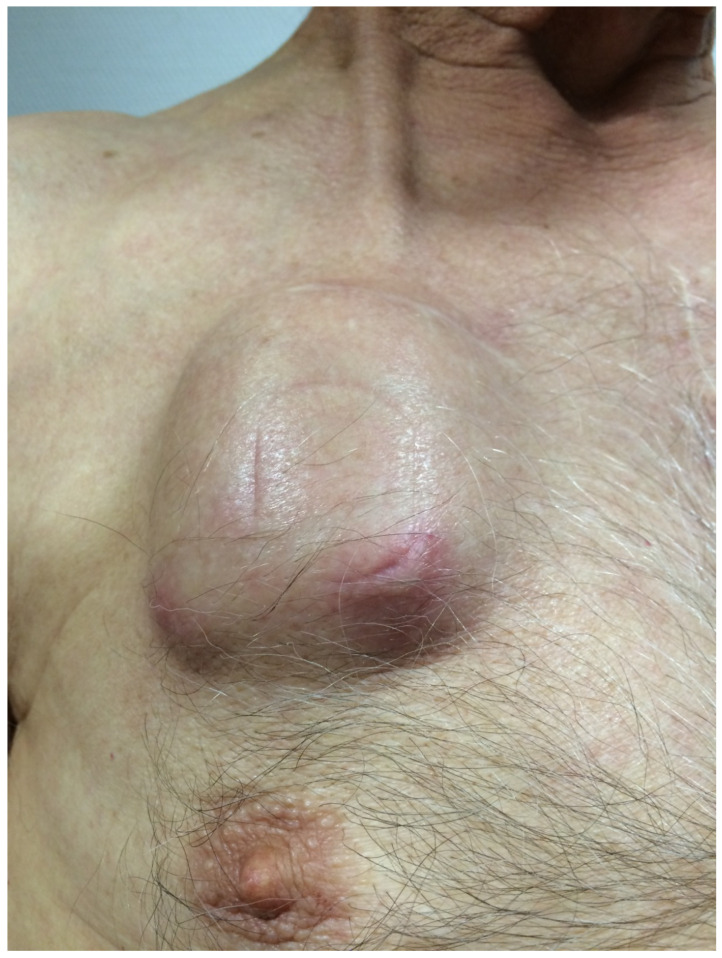
Patient 4 presented with a swollen generator pocket several weeks after an infected bee sting directly above the generator.

**Figure 5 brainsci-13-01650-f005:**
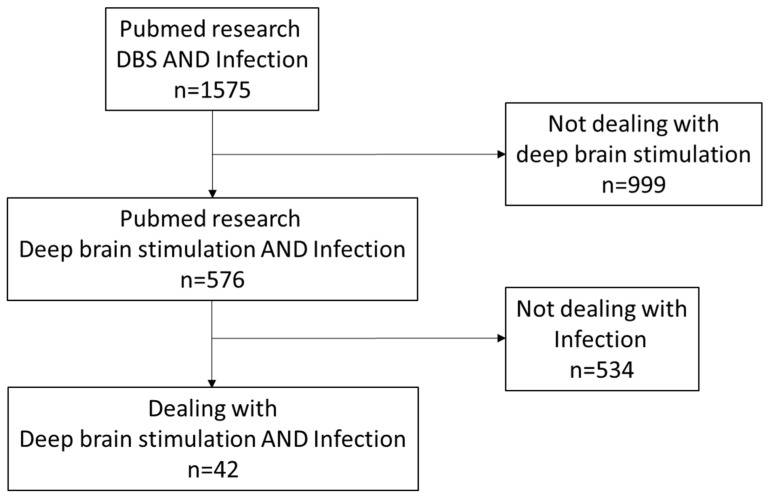
PubMed research within a PRISMA flow diagram leading up to our cited articles.

**Table 1 brainsci-13-01650-t001:** Patients’ Characteristics: The mean age of the patients was 81.6 years, and all suffered from essential tremor (ET). The initially implanted DBS system lasted an average of 272 days (ranging from 13 to 805). Half of the patients had an identifiable pathogen causing the infection. Patient 2 experienced a recurrent infection 18 months after the salvage surgery. All patients except for Patient 4 were obese, with a mean body mass index (BMI) of 30 kg/m^2^.

Patient	1	2	3	4	5	6	Mean
Age	89	81	73	86	83	78	81.7
Sex	Female	Male	Male	Male	Female	Female	
Disease	ET	ET	ET	ET	ET	ET	
Days between last and salvage surgery	65	377	336	805	13	36	272
Body mass index	36.7	29.4	32.3	20.9	35.6	24.8	30.0
Causing pathogen	Yes	Yes	Yes	No	No	No	
Months without complication	29	18	89	74	71	67	58.0

**Table 2 brainsci-13-01650-t002:** All patients had additional comorbidities, including arterial hypertension (AHT), coronary artery disease (CAD), cholecystectomy (CCY), non-insulin dependent diabetes mellitus type 2 (NIDDM), dilatative cardiomyopathy (CM), obstructive sleep apnea syndrome (OSAS), or chronic obstructive pulmonary disease (COPD). Three patients had normal C-reactive protein (CRP) levels and all had normal leukocyte counts. Pathogens were found in half of the patients, and except for *Proteus mirabilis*, they were part of the normal skin flora. IV antibiotics (mainly cefuroxime) were administered for an average of 8.5 days and continued orally for 6 days.

Patient	Age	Comorbidity	Smoker	CRP (mg/dL)	Leukocyte Count (n/µL)	Pathogen	IV Antibiotic	Oral Antibiotic
1	89	AHT	No	9.5	normal	*S. aureus*	Cefuroxime (12 days)	Flucloxacillin (6 days)
2	81	AHT, hypothyroidism	No	4.6	normal	*Proteus mirabilis*	Cefuroxime (4 days)	none
3	73	AHT, CAD with bypasses, atrial fibrillation, aortic valve stenosis, bicuspid valve stenosis, OSAS	Yes	<0.5	14.1 (smoker)	*S. aureus S. epidermidis*	Cefuroxime (11 days)	Cotrimoxazol (3 days)
4	86	AHT, CAD with bypasses, xenogeneic aortic valve replacement	No	<0.5	normal	none	Cefuroxime (4 days)	none
5	83	Dementia, CCY, NIDDM, hypothyroidism	No	<0.5	normal	none	Cefuroxime (6 days)	Cefuroxime (8 days)
6	78	AHT, dilatative CM, COPD, breast cancer	No	5.3	normal	none	Flucloxacillin (14 days)	Rifampicin (7 days)
Mean	81.7						8.5 days	6 days

## Data Availability

Data are contained within the article.

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
