# Peer review of "Rescuing Infected Deep Brain Stimulation Therapies in Severely Affected Patients"

_brainsci, 2023, doi:10.3390/brainsci13121650_

Round 1

Reviewer 1 Report

Comments and Suggestions for Authors

Comments to the authors:

- ABSTRACT and INTRODUCTION: The infection of deep brain stimulation (DBS) hardware could not be considered as a “common” complication in DBS surgeries but rather only a possible one.

- In my opinion the description of the different cases should be moved from the Materials and Methods section to the Results Section. To date the structure of the manuscript appears a little confusing. In order to reconsider the article for a possible publication it will be necessary to significantly improve the description of the single cases.

- In addition the authors should better specify the review of the literature that they have performed moving that paragraph in the Materials and Methods and Results sections.

I also think that this manuscript should include a detailed revision of the literature with the corresponding PRISMA flow diagram and the detailed description of the search strategy in order to be considered as a case description and review of the literature article.

Comments on the Quality of English Language

Moderate editing of English language required

Reviewer 2 Report

Comments and Suggestions for Authors

This method likely has been implemented in some facilities, yet no documented evidence of such implementation exists thus far. Therefore, it is imperative to thoroughly document this method. I have some suggestions and questions.

1. The guidelines for the management of device-related infections recommend the administration of vancomycin as the first-line antibiotic. However, authors (as well as other clinicians) who have experienced device-related infection in the context of DBS tend to prefer cephalosporin antibiotics of 2nd or 3rd generations as the initial treatment choice. The rationale for not using vancomycin first should be discussed.

2. The highlight of this paper lies in the method for reconstructing DBS devices. To enhance the reader's comprehension of this method, it is suggested that the authors include illustrative drawings depicting the surgical procedures involved.

3. The content of the tables is somehow redundant. For example, all the tables feature repeated information such as age and gender. Tables 2 and 3 can be amalgamated into a single table.

4. The infection rate of DBS devices appears to be relatively high in comparison to other medical devices. It would be valuable for the authors to show the generally reported percentage of infection from several references in this field and conduct a comprehensive discussion on this matter.

5. Typo error at line 36. "This true" should be "This is true". 

Reviewer 3 Report

Comments and Suggestions for Authors

The paper discusses a novel technique for managing infections in deep brain stimulation (DBS) hardware, aiming to provide an alternative to hardware removal to prevent patient dependency. While the idea is intriguing, the paper requires significant revisions and technical clarifications to enhance its scientific rigor and readability.

1. Expand the introduction to provide a comprehensive overview of the issue of DBS hardware infections, their prevalence, and current management strategies.

2. Clearly state the objectives and research questions addressed by the novel technique.

3. Provide a detailed and step-by-step description of the novel surgical technique, including the placement of new electrodes, extensions, and generators on the contralateral side.

4. Explain the criteria for patient selection and inclusion in the case series.

5. Specify the surgical team's expertise and qualifications in performing these procedures.

6. Include statistical analyses if applicable.

Clarify whether laboratory alterations and pathogens were found in the other half of the patients or if no pathogens were identified at all.

7. Interpret the results in the context of the current literature on DBS hardware infections. Discuss the advantages and limitations of the proposed technique compared to hardware removal. Address potential complications or risks associated with the contralateral approach.

8. Cite relevant literature to support the rationale for the novel technique and provide context for the study.

Comments on the Quality of English Language

Fine to me

Round 2

Reviewer 3 Report

Comments and Suggestions for Authors

Thank you for incorporating the changes. I can now recommend acceptance of this manuscript in its current form (subject to fulfillment of other reviewers' suggestions).